# Periodic Variations of Solar Corona Index during 1939–2020

**Rui Tang [1], Yu Fei [1,*], Chun Li [1], Wen Liu [2], Xinan Tian [3] and Zhongjie Wan [3]**

1   School of Statistics and Mathematics, Yunnan University of Finance and Economics, Kunming 650221, China; tangrui3@stu.ynufe.edu.cn (R.T.); lichun3@stu.ynufe.edu.cn (C.L.)
2   College of Science, Yunnan Agriculture University, Kunming 650201, China; liuwen@ynau.edu.cn
3   Chongqing University of Arts and Sciences, Chongqing 402160, China; tianstu@cqwu.edu.cn (X.T.); wan_zj@cqwu.edu.cn (Z.W.)
*   Correspondence: feiyu@ynufe.edu.cn

**Abstract:** Periodic behaviors of solar magnetic indicators might provide a clue for the understanding of solar dynamic processes. Combining with a Lomb–Scargle periodogram, the concentration of frequency and time via a multitapered synchrosqueezed transform is applied to investigate the periodic variations of modified coronal index for the time interval from 1 January 1939 to 31 August 2020. The main results are as follows: (1) During solar cycles 19 to 23, the Schwabe cycle of the modified coronal index is operating with its length variating between 10.5 and 11-yr, and the average value of length is 10.67-yr with standard deviation of 0.14-yr. (2) The Rieger-type periods are mainly distributed in a range from 120 to 200 days. In addition, the periods vary somewhat intermittently during cycles 18 to 24, which are operating with the highest power in cycles 21 and 22 while the power is much lower in cycles 23 and 24. (3) For rotation periods, the temporal variation exhibits a highly intermittent pattern as an asymmetrical distribution with its 25th, 50th, and 75th quantile of 26, 27.8, and 31-day, respectively. (4) Other mid-range periods are also detected with an average period length of 8.07, 5.44, 3.42, 2.3, and 1.01-yr.

**Keywords:** solar activity; corona; periodicity; Lomb–Scargle periodogram; ConceFT





## 1. Introduction

Cyclic variability is one of the most interesting and important phenomena of solar cycles. A search for periodicities in solar activity indicators (such as the sunspot numbers/areas, Hα flare index, coronal index, and so on) is an important aspect in solar physics fields, since any detection of periodicity might provide a clue to understanding the dynamics mechanism of solar activity [1]. Periods of 11 years (11-yr) and 27 days (27-day) have been well-established in almost all of the solar indicators. The former is related to the solar magnetic activity cycle [2], and the latter reflects the modulation imposed by the solar rotation [3]. Another famous long-term periodicity is the 22-yr magnetic cycle [4], which has been named the Hale cycle.

Many other periods with different time scales have been found in various indicators. For example, a periodicity of 154-day was discovered by Rieger et al. during γ-ray flare occurrences in solar cycle 21 [5], and since then, lots of authors searched for the existence of "mid-range" periods (lying between 27 days and 11 years [6,7]) in various indicators of the Sun. Pap [8] and Pap et al. [9] found a 23.5-day period in solar irradiance. During solar cycles 19–24, various fluctuations with "mid-range" periods have been detected by authors in different phases of different solar cycles, such as 51, 62, 73, 152, and 546-day [10–12]. A type of oscillation with periods of around two years, which have been named qusi-biennial oscillations (QBOs) of the Sun, have also been found in many indicators, and have been paid increasing attention by authors in recent decades [13–16].

It is well known that the solar magnetic activity is governed by a complex dynamo mechanism and exhibits a nonlinear dissipation behavior in nature [17,18]. For a long time,

the complex temporal and spatial behaviors of solar corona were studied continuously in various ways by many authors. Leroy and Noens [19] examined the latitude distribution variation of the coronal magnetic activity during the time span from 1944 to 1974, and found that the whole evolution process of the magnetic activity occurring in the corona spreads over a time interval of around 17 years, which is longer than the time interval (11-yr) between two consecutive solar cycles. By investigating the homogeneous time series of coronal green line intensities for the time interval from 1964 to 1990, Rybák [20] calculated the averaged synodic rotation period. They found that the coronal rotation period for the whole range of latitudes is 28.18 ± 0.12 days and the rotation period in latitudinal band ±30° is 27.65 ± 0.13 days. By studying the observational data of the brightness of the coronal green line (530.3 nm) and sunspot activity indices over the period of 1939–2001, Badalyan et al. [21] found that solar QBOs in the asymmetry time series of activity indicators are more obvious than the QBOs that exist in the activity indicators themselves. Robbrecht et al. [22] found that the latitude diagram of coronal emission exhibits an enhanced brightness zone, which occurs close to the two poles after the maximum time of a solar cycle.

Using the daily series of coronal index in solar cycle 23, Chowdhury and Dwivedi [23] detected a prominent period of 22 to 35 days in the high-frequency range, the Rieger period of 150 to 160-day during different phases of cycle 23, and a number of quasi-periodic oscillations. Deng et al. [24] investigated the phase asynchrony between coronal index and sunspot numbers. They pointed out that the coronal index and sunspot numbers are coherent in low-frequency components corresponding to the 11-yr Schwabe cycle, but they are asynchronous in phase in high-frequency components, namely the sunspot numbers begin one month earlier than the coronal index. By investigating the series of coronal global rotation in the 10.7-cm solar radio flux (2800 MHz), Xie et al. [25] found a typical periodicity of 6.6-year of the coronal rotation, which was interpreted as the third harmonic of the Hale cycle (22-yr). Mancuso et al. [26] studied the intermediate periodicities of the coronal green emission line (530.3 nm) for the time interval from 1944 to 2008. They pointed that the uneven latitudinal distribution of solar QBOs could be considered as a fundamental, but puzzling, characteristic of solar magnetic activity. Kilcik et al. [12] compared the temporal and periodic variations of the Maximum CME Speed Index (MCMESI) and the number of different class solar X-ray flares for Solar cycles 23, 24. They detected periodicity of 546-day in MCMESI, and found that all the X-ray solar flare classes show remarkable positive correlation with the MCMESI, and all class flare numbers and the MCMESI show similar periodic behavior. Deng et al. [27] used the series of modified coronal index for the date from 1 January 1939 to 31 May 2019, to investigate the systematic regularities of solar coronal rotation. They found that the period of 27.5-day is the only synodic coronal rotation period shorter than 64 days, and significant periods of 3.25, 6.13, 9.53 and 11.13-yr exist in coronal rotation.

With the hope of enriching further information on the temporal variations and underlying processes of solar coronal activity, we adopt a Lomb–Scargle periodogram and a relatively novel method—concentration of frequency and time via a multitapered synchrosqueezed transform (ConceFT) [28], to detect the periodicities in a modified coronal index (MCI) [29]. A brief introduction to the data and the methods used in this paper are given in the next section. time–frequency representations of signal components and the statistical results are given in Section 3. Finally, the conclusions are presented in Section 4.

## 2. Data and Methods

### 2.1. The Modified Coronal Index

The coronal index (CI) is one of the most typical indicators in the corona of the Sun, which originally has been constructed on the basis of ground-based measurements for the intensities of coronal green radiation from the Fe XIV emission line at 530.3 nm. The daily value of this index have been recorded since 1 January 1939 [30–32]. From a Sun observing

space-based probe, Luká and Rybanský [29] introduced the modified coronal index (MCI) to replace the original coronal index, which can replace existing CI in all uses and contexts.

The data series of MCI is downloaded from the website of the Slovak Central Observatory in Hurbanovo (http://www.suh.sk/obs/vysl/MCI.htm, accessed on 15 June 2022). Figure 1 shows a daily time series of a modified coronal index for time interval from 1 January 1939–31 August 2020, which nearly covers the solar cycles 17–24.

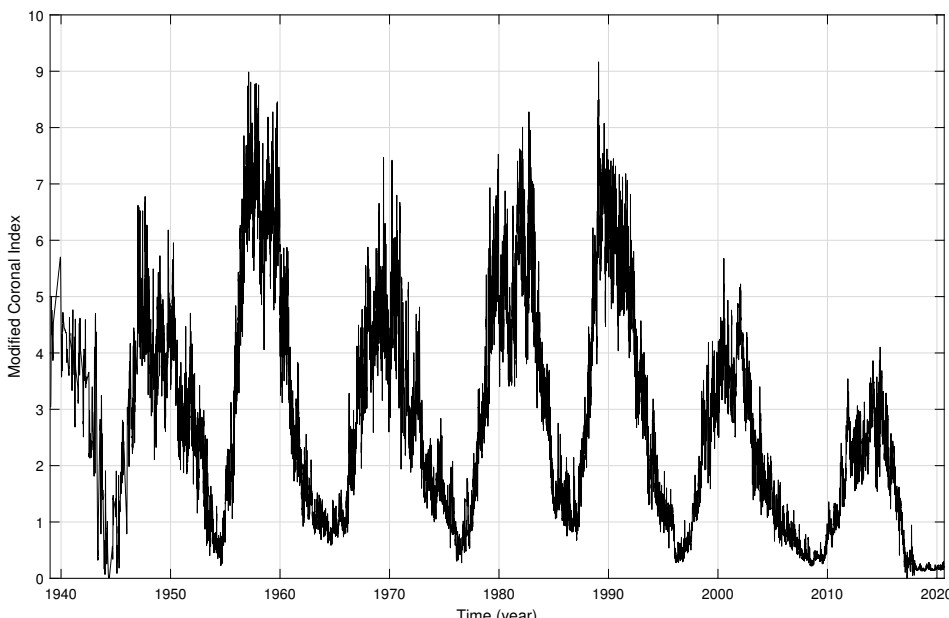

**Figure 1.** Daily time series of modified coronal index for 1 January 1939–31 August 2020.

*2.2. Lomb–Scargle Periodogram*

The Lomb–Scargle periodogram is a typical method to analyze the power spectra of a time series, which was first proposed by Lomb [33] and then developed by other authors, such as Scargle [34] and Horne et al. [35]. This method allows for efficient computation of a power spectrum estimator to find and test weak periodic signals in otherwise random, unevenly sampled data. Therefore, it is wildly applied to calculate the periodicities of data in astronomic fields; for example, several authors used it to investigate the periodic behaviors of solar magnetic activity indices [36,37].

For a time series $x(t_k), k = 1, \cdots, N$, the Lomb–Scargle periodogram is defined by

$$P(f) = \frac{1}{2\sigma^2} \left\{ \frac{\left[\sum_{k=1}^{N}(x_k - \overline{x})cos(2\pi f(t_k - \tau))\right]^2}{\sum_{k=1}^{N} cos^2(2\pi f(t_k - \tau))} + \frac{\left[\sum_{k=1}^{N}(x_k - \overline{x})sin(2\pi f(t_k - \tau))\right]^2}{\sum_{k=1}^{N} sin^2(2\pi f(t_k - \tau))} \right\}, \quad (1)$$

where $\overline{x} = \frac{1}{N}\sum_{k=1}^{N} x_k$ and $\sigma^2 = \frac{1}{N-1}\sum_{k=1}^{N}(x_k - \overline{x})^2$. The parameter $\tau$ is determined by equation

$$tan(2(2\pi f)\tau) = \frac{\sum_{k=1}^{N} sin(2(2\pi f)t_k)}{\sum_{k=1}^{N} cos(2(2\pi f)t_k)}. \quad (2)$$

If the input signal consists of white Gaussian noise, then $P(f)$ follows an exponential probability distribution. Based on the distribution, one can test whether a peak comes from signal or noise. Namely, for a given peak in the periodogram, the statistical significance is estimated by computing the false alarm probability (FAP) by the equation

$$FAP(z_k) = 1 - [1 - exp(-z_k)]^N, \quad (3)$$

where $z_k$ is the height of the peak in the normalized power spectrum [34,35]. Thus, the statistical significance is $(1 - FAP(z_k))$.

### 2.3. Wavelet Transform

The wavelet transform is a classical and primary method that transforms signals from time domain to time-scale (time–frequency) domain. It is widely used in signal analysis because it provides more detailed information of both frequency and time position. For a continuous time signal $S(t)$, a $\psi$-based wavelet transform $WT_\psi$ at $(a,b)$ is defined as

$$WT_\psi(a,b) = \frac{1}{\sqrt{a}} \int S(t)\psi^*(\frac{t-b}{a})dt, \qquad (4)$$

where $\psi$ is the mother wavelet, and $\psi^*$ denotes its complex conjugate; $a$ is a dilated scale that is related to the physical frequency, and $b$ represents time position. One could reconstruct the original signal by

$$S(t) = C_\psi^{-1} \int \int \frac{1}{a^2} WT_\psi(a,b)\psi(\frac{t-b}{a})dadb, \qquad (5)$$

where $C_\psi$ is a normalization factor that only depends on the $\psi$.

### 2.4. Synchrosqueezed Wavelet Transform

The synchrosqueezed wavelet transform (SWT) is a tool developed for extracting time–frequency information of a signal that was firstly introduced by Daubechies et al. [38]. It has been widely used in various scientific fields by many authors in recent years [16,27,39,40]. By squeezing and rearranging the energy at a center frequency in the spectrum from wavelet transformation, a new spectrum with more concentrated energy on the time–frequency surface can be obtained. Compared to the classical wavelet transform, the SWT improves the resolution of in the time–frequency diagram. For a signal $S(t)$, the steps of the SWT are as follows:

(1) Performing the wavelet transform on $S(t)$, to obtain wavelet coefficient $WT(a,b)$;

(2) Calculating the corresponding candidate frequency by

$$\omega(a,b) = -i(WT(a,b))^{-1}\frac{\partial WT(a,b)}{\partial b}, \qquad (6)$$

where $i$ presents the imaginary unit;

(3) For a center frequency $\omega_l$, the synchrosqueezing wavelet transform $T_S$ at $(\omega_l, b)$ is defined as

$$T_S(\omega_l, b) = (\Delta\omega)^{-1} \sum_{a_k:|\omega(a_k,b)-\omega_l|\leq\Delta\omega/2} W_s(a_k,b)a_k^{-3/2}\Delta a_k, \qquad (7)$$

where $a_k$s presents the discrete scales with a scale step $\Delta a_k = a_k - a_{k-1}$; $\Delta\omega/2$ denotes a half band width around the center frequency $\omega_l$, namely, $\Delta\omega = \omega_l - \omega_{l-1}$.

The inverse transformation of the synchrosqueezing wavelet transform, which can be used to reconstruct time-domain signal, is defined as

$$S(t) = Re\left\{C_\psi^{-1} \sum_l T_S(\omega_l, t)\Delta\omega\right\}, \qquad (8)$$

where $C_\psi$ is a normalization constant that depends on the $\psi$, and $Re$ denotes the real part of a complex variable.

### 2.5. CWT-Based ConceFT

Concentration of frequency and time via a multitapered synchrosqueezed transform (ConceFT) is a relatively novel method of time–frequency representation, which was proposed by Daubechies et al. [28]. The ConceFT uses several orthonormal tapers, and averages the reassigned time–frequency representations determined by each of the individual tapers. Therefore, the concentration for a "true" signal component will be in similar locations in

the time–frequency plane for each of individual representations, whereas the spurious concentrations, artifacts of correlations between noise and the windowing function, tend not to be co-located and have a diminished impact when averaged. An algorithm of ConceFT based on continuous wavelet transform (CWT-based ConceFT) is performed as the following steps:

(1) Take $I$ orthonormal tapers, $\psi_1(t), \ldots, \psi_I(t)$;

(2) Pick $N$ random vectors $r_n, n = 1, \ldots, N$, of unit norm in $\mathbb{R}^I$;

(3) For each $n = 1, \ldots, N$, define $\psi_{[n]} = \sum_{i=1}^{I} (r_n)_i \psi_i(t)$;

(4) For each $n = 1, \ldots, N$, take $\psi_{[n]}$ as the taper, perform the SWT on signal $S(t)$, to obtain its reassignment coefficient $T_s^{(\psi_{[n]})}(\omega_l, b)$;

(5) The final ConceFT representation of $S(t)$ at $(\omega_l, b)$ is then the average

$$CFT_s(\omega_l, b) = \frac{1}{N} \sum_{n=1}^{N} T_s^{(\psi_{[n]})}(\omega_l, b). \tag{9}$$

In practice, $I$ could be as small as 2, while $N$ could be chosen as large as the user wishes.

## 3. Results and Discussion

In this work, we firstly identify the significant periods in MCI using the Lomb–Scargle periodogram. After that, the ConceFT method is executed on the series of MCI to explore the temporal variation patterns of these prominent periods.

### 3.1. Lomb–Scargle Periodogram of MCI

The Lomb–Scargle periodogram of MCI can be seen in Figure 2. One could find that the 10.22-yr period is the most prominent period, which is related to the Schwabe solar cycle (~11 years). Such period was obvious from the observational data and has been commonly well established in other solar magnetic activity indices. Another significant period with a higher statistical confidence level is 8.17-yr cycle. Its period length is close to the Schwabe period, but the normalized power is much lower. Thus, they could be driven by different physical mechanisms and further study is required.

There are many other periods where the peaks are higher than the 99% significance level. The periods of 4.54, 5.45, and 6.29-yr are close to half of the Schwabe cycle. It may be the second harmonic of the 10.22-yr Schwabe cycle, since their mean value of 5.42 is nearly 5.11-yr (10.22/2 = 5.11). In other words, such periods may be induced by the higher order beat of the Schwabe cycle. Some authors argued that such periods may be the 4th harmonic of the 22-yr magnetic cycle because 5.42-yr is closed to 5.5-yr (22/4 = 5.5) [27].

Periods around two years can be interpreted as the quasi-biennial oscillations (QBOs). It has been paid more attention by authors recently, since its physical origin may be related to the dynamic process in the solar tachocline. Some authors studied the QBOs during the period range of 1.5 to 3-yr, but others extend the range from 1.5 to 4-yr. [13,14,16,27]. However, from the lower left panel in Figure 2, one can see that the periods around 3.5-yr have much more energy than that around 2.5-yr. Thus, the two classes of periods are more likely driven by different mechanisms. The average period of 3.27, 3.55, and 3.89-yr is 3.57-yr. These periods may be the third harmonic of the 10.22-yr cycle, since the value 3.57-yr is nearly 3.41-yr (10.22/3 = 3.41). The periods 1.95, 2.27, 2.48, and 2.92-yr can be taken as quasi-biennial oscillations.

For the 0.98-yr cycle (~1-yr), its exact physical mechanism is still in doubt, since it is hard to rule out the possibility of seasonal effects of the Earth. Thus, it is regarded as mostly relating to the annual-variation signal [27]. From Figure 2, we could not find any significant periods of the 99% level range from 2 to 200 days. However, many authors focused on the periods within such an interval, since they have exact physical meaning. For example, the period of its length around 27-day may arise from solar rotation, and the Rieger type periods about 153-day may be a property of merging flux on the solar disk [41].

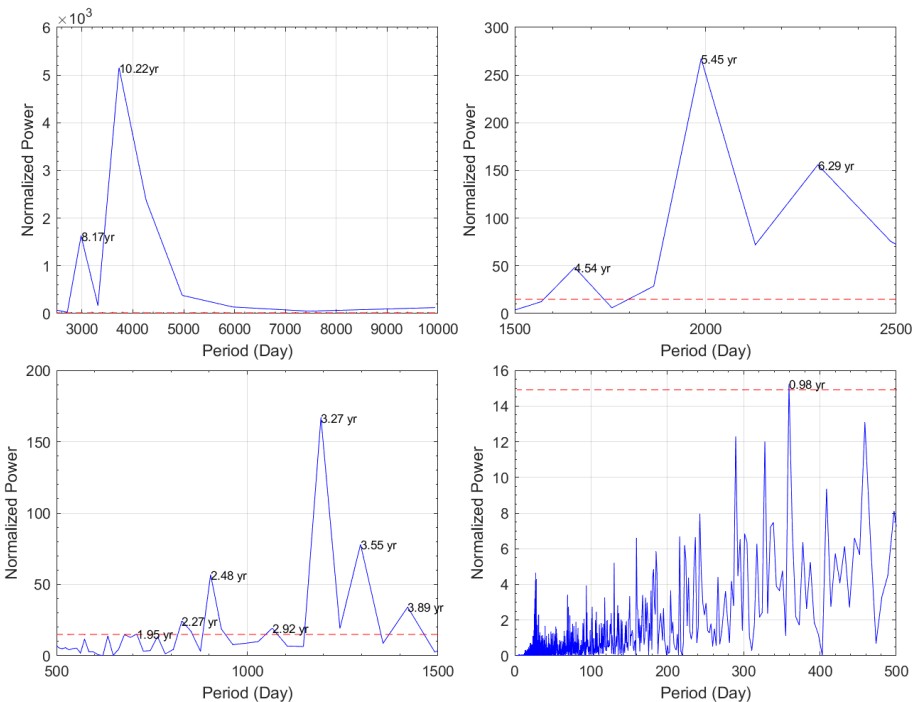

**Figure 2.** Lomb–Scargle periodogram of the modified coronal index. The red dashed lines are the 99% significant level. (**Upper left panel**): periods from 2500–10,000 days; (**Upper right panel**): periods from 1500–2500 days; (**Lower left panel**): periods from 500–1500 days; (**Lower right panel**): periods from 2–500 days.

### 3.2. Variation Patterns of Significant Periods

In general, the significant periods found in the Lomb–Scargle periodogram are not constant during the considered time interval due to the complexity of solar activity. Therefore, the ConceFT method is performed on the series of MCI to detect the temporal variation patterns of these periods for revealing further information.

### 3.2.1. The Schwabe Cycle

For comparing to the traditional continuous wavelet transform (CWT), Figure 3 shows the time–frequency planes which are derived from both methods with the period scales between 1 and 20-yr. From Figure 3, one can see that the Schwabe cycle (~11-yr) is the most prominent period in both planes, which is operating with higher power in the whole duration of the time being considered. Furthermore, the ConceFT exhibits higher resolution in its time–frequency plane than CWT, which means that the ConceFT is more powerful to concentrate the energy of a signal and is more capable of diminishing the noise.

However, notice that the prolongation of the periodicity during cycles 23–24 exhibits the opposite way between CWT and ConceFT. For explanation, we could examine the observed data in normal solar cycles shown in Figure 4. From the average length of normal solar cycle based on observed data shown in Figure 4, we can find that the cycle length increases from 9.92 years in cycle 22 to 12.33 years of cycle 23, and then decreases to 11 years for cycle 24. The ConceFT has revealed such a characteristic while the CWT exhibits the opposite way in cycle 24. From this viewpoint, the ConceFT has more ability to capture the detailed features hidden in data. However, from Figure 3, we can see that the prolongation of the period has exceeded the lines of cone of influence (COI) during 2005–2020 and 1939–1955, and begins to bifurcate. Hence, the edge effects may distort the pattern in both methods.

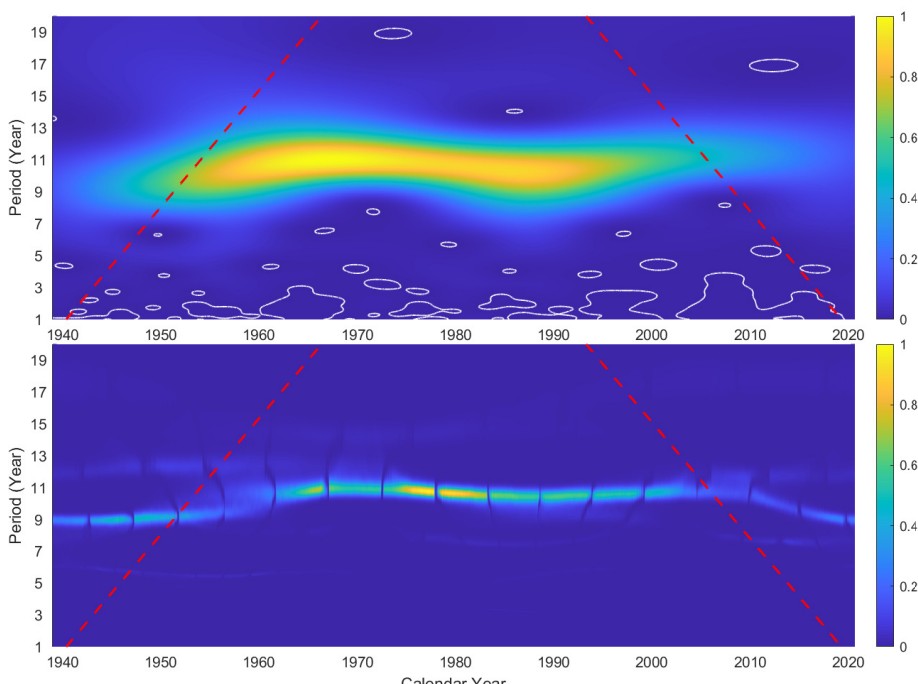

**Figure 3.** Time–frequency representation of the modified coronal index series with periods between 1 and 20-yr. (**Upper panel**): derived from the continuous wavelet transform [42]; the dashed red line is the cone of influence (COI); the solid white line is the 99% confidence level; (**Lower panel**): derived from the CWT-based ConceFT.

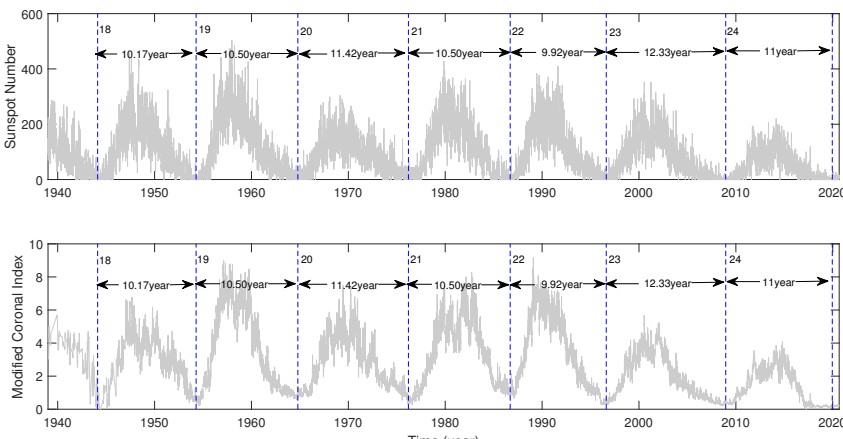

**Figure 4.** Observed data of the sunspot numbers and the modified coronal index in normal solar cycles 18–24; Upper panel: sunspot numbers; Lower panel: modified coronal index.

In order to verify the edge effects and the effectiveness of the new method, we could perform the ConceFT on the series of sunspot numbers (SSN) with different length. The data can be downloaded from the website http://www.sidc.be/silso/datafiles (accessed on 15 June 2022). Figure 5 illustrates the time–frequency representations of sunspot numbers series derived by ConceFT. The upper panel is of a longer series from 1 January 1900 to 31 August 2020. The lower panel is of a shorter series from 1 January 1939 to 31 August 2020, for which the observed time interval is the same as the series of the modified coronal index. It can be seen that the duration from 1955 to 2005 is in the region of the COI in both series. The variation patterns of the Schwabe cycle in both series are in accordance with each other. Namely, it is rising during 1955–1975, then declining in the time interval 1975–1990, and,

from 1990, it is increasing again. It fits well with the observed data shown in the upper panel of Figure 4.

Nevertheless, the durations that are out of the COI are doubtful. For the longer series (upper panel), the duration from 1939 to 1955 is not out of the COI for a period of 11-yr, and the periodicity shows a single smooth variation pattern of values between 9.5 and 12-yr. However, for the shorter series (lower panel), the duration from 1939 to 1955 has been out of the COI where the energy distribution becomes vague and bifurcated. In addition, during 2005–2020, where it has been out of the COI for such period in both series, the prolongation of the periodicity is operating in different directions. Therefore, the ConceFT method is still restrained by edge effects, but it works well in the duration that has no "edge effects".

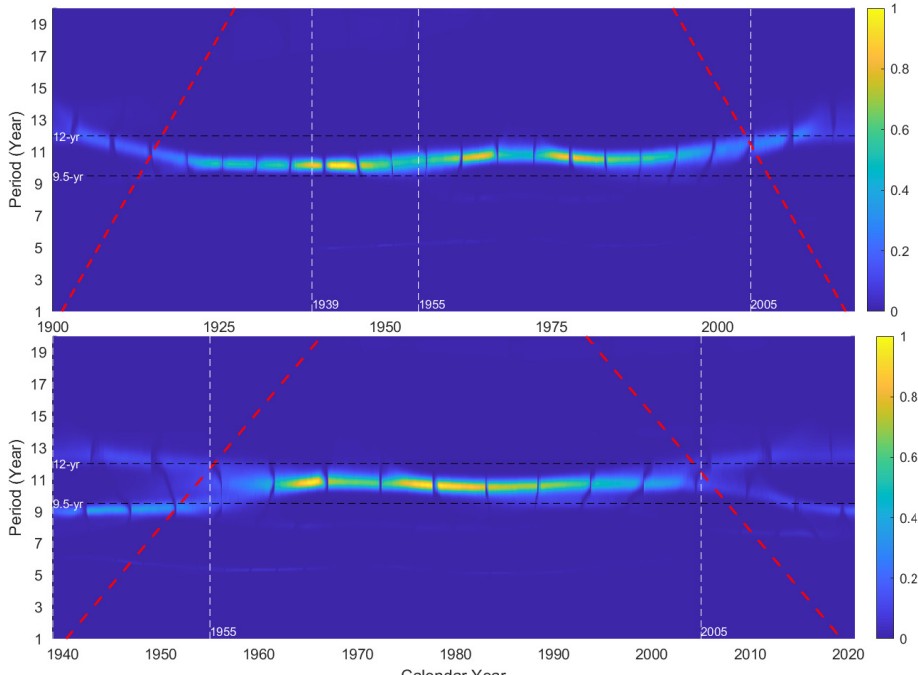

**Figure 5.** Time–frequency representation of the sunspot numbers data with different length of series. The dashed red lines are the cone of influence (COI). (**Upper panel**): series from 1 January 1900 to 31 August 2020; (**Lower panel**): series from August 1939 to 31 August 2020.

The Schwabe cycle is the most famous and important cycle that exists in all solar magnetic indicators. However, its detailed patterns of variation in various indicators are different from each other. To detect the pattern of temporal evolution for the Schwabe cycle of the modified coronal index in more detail, the time–frequency plane on timescales from 8.5 to 13.5-yr along with temporal evolution profile of Schwabe cycle are showed in Figure 6. The spectrum obtained by ConceFT is given in the upper panel. The period variation profile, which is performed with maximum power at each time point, is shown in the lower panel.

From the lower panel of Figure 6, we can see that the Schwabe cycle is operating with the values between 10.5 and 11-yr during the time interval from the maximum times of solar cycle 19 to the end time of solar cycle 23. Meanwhile, using the period values from the lower panel, one can calculate the average period length of 10.67-yr with the standard deviation of 0.14-yr. During cycles 19 and 20, the period length is rising, then declining in cycle 21 and 22, and increasing again during cycle 23. This pattern is in accordance with the observed data shown in Figure 4.

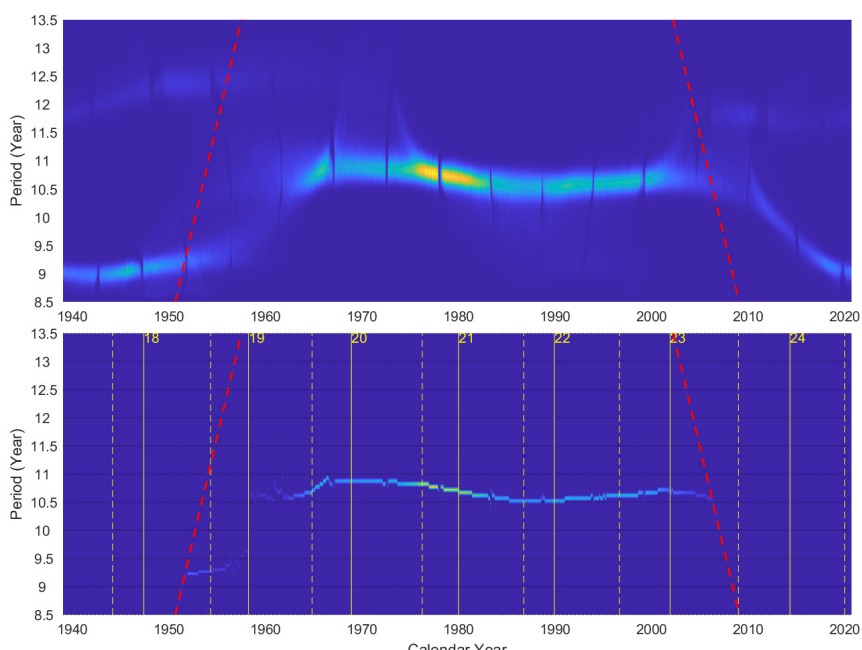

**Figure 6.** Time–frequency representation of the modified coronal index series with a period between 8.5 and 13.5-yr.The dashed red lines are the cone of influence (COI). (**Upper panel**): the spectrum obtained by ConceFT; (**Lower panel**): the profile of period variation performed by the maximum power at each time point. The vertical solid and dashed yellow lines in the lower panel indicate the maximum and minimum times of normal solar cycles, respectively.

### 3.2.2. The Other Significant Periods

Figure 7 illustrates the ConceFT time–frequency plane with scales ranging from 6.5 to 8.5-yr. The average value of periods is 8.07-yr with a standard deviation of 0.38-yr. During cycle 21, the period declines from 8.1-yr to 7.5-yr. In cycle 22, the period increases from 7.5-yr to 8.3-yr. From the maximum times to the end times of cycle 23, such period once again decreases from 8.5-yr to 7.6-yr.

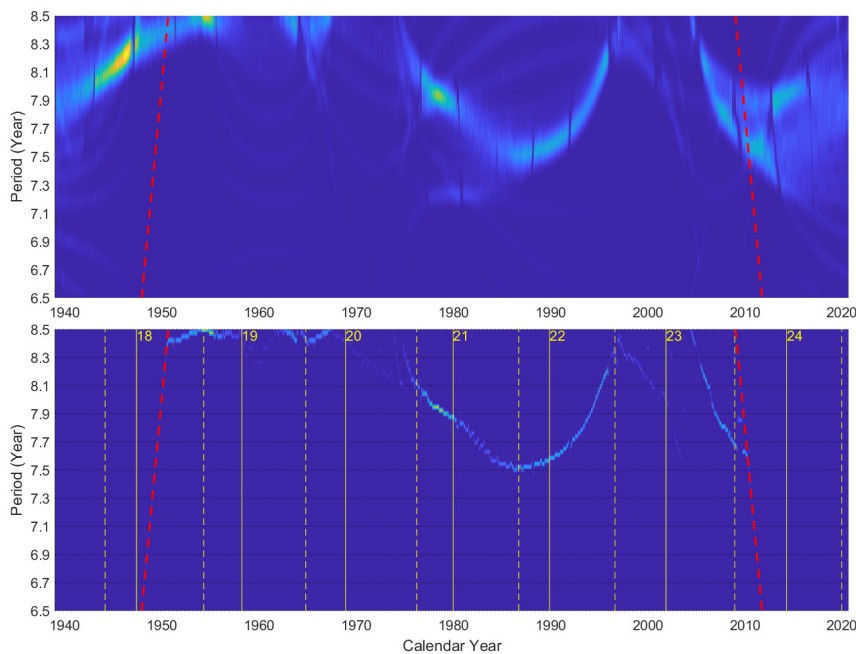

**Figure 7.** Same as Figure 6, but for periods between 6.5 and 8.5-yr.

Figure 8 demonstrates that the time–frequency representation for scales ranges from 4-yr to 6.5-yr, which displays a continuous variation pattern. A trend of its period length decreases from 5.8-yr to 5.5-yr in the duration from the maximum times of cycle 18 to the maximum times of cycle 19; and for duration from the maximum times of cycle 19 to maximum times of cycle 20, the period of its length increases from 5.5-yr up to 5.8-yr. In cycles 21–23, such prominent period increases from 4.8-yr to 6.3-yr. According to the time scale ranges from 4 to 6.5-yr, the average period is calculated as 5.44-yr with a standard deviation of 0.44-yr.

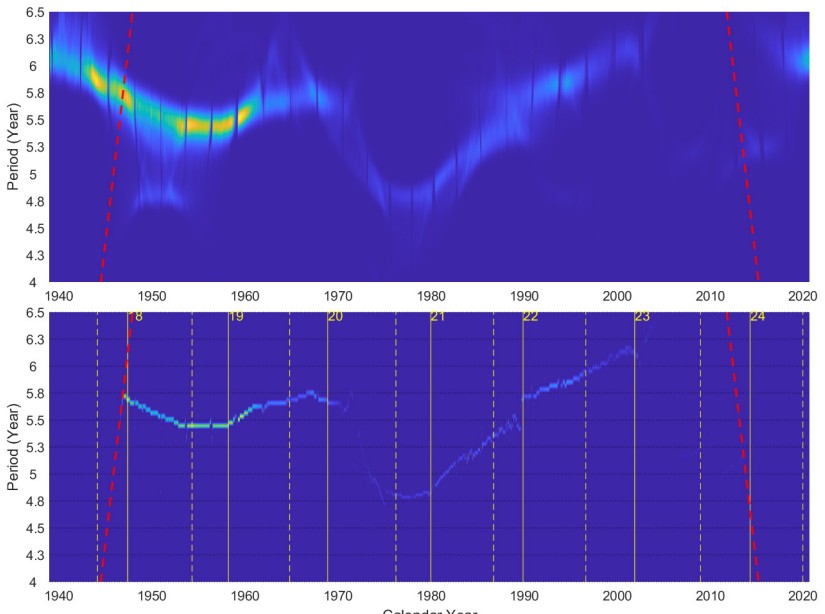

**Figure 8.** Same as Figure 6, but for periods between 4-yr and 6.5-yr.

As can be seen from Figure 9, periods are continuously operating from the beginning of solar cycle 21 to the end time of cycle 22 with much higher energy, while they appear somewhat intermittently in other times with lower power. The average value of periods is 3.42-yr with a standard deviation of 0.24-yr in a time scale ranging from 3 to 4-yr.

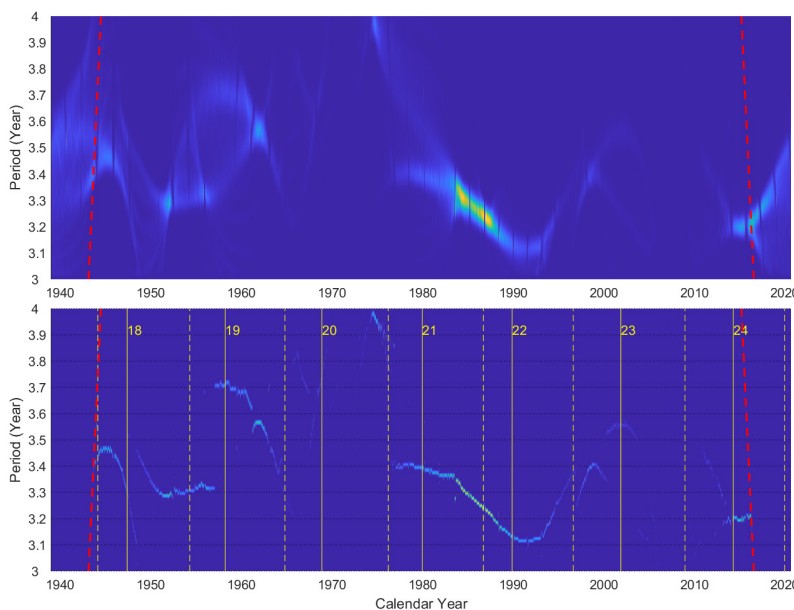

**Figure 9.** Same as Figure 6, but for periods between 3 and 4-yr.

Figure 10 shows the time–frequency plane for periods ranging from 1.5 to 3-yr. Many authors take the periods lying in such range as QBOs. As can be seen in Figure 10, the periods are predominantly distributed in the range from 2 to 3-yr. These periods appear somewhat intermittently during cycles 18–24 with more energy in cycles 18, 19, and 21. The average value of periods is 2.3-yr with a standard deviation of 0.42-yr in time scale ranges from 1.5 to 3-yr.

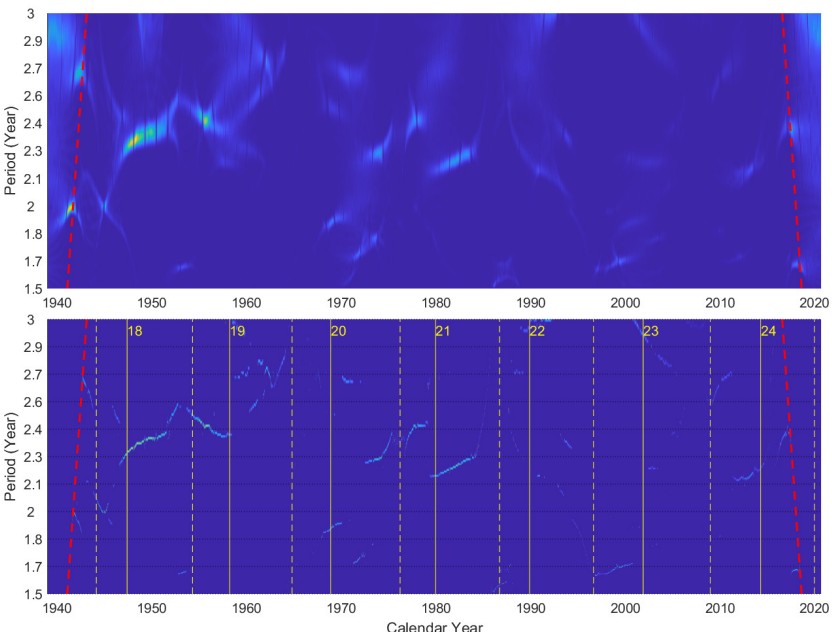

**Figure 10.** Same as Figure 6, but for periods between 1.5 and 3-yr.

For time scale ranges from 0.5 to 1.5-yr, Figure 11 shows the time–frequency plane and the period variation profile. It can be seen that these periods are operating somewhat intermittently with higher power in cycles 19–22 while their power is much lower in other solar cycles. The average period is 1.01-yr and the standard deviation is 0.24-yr.

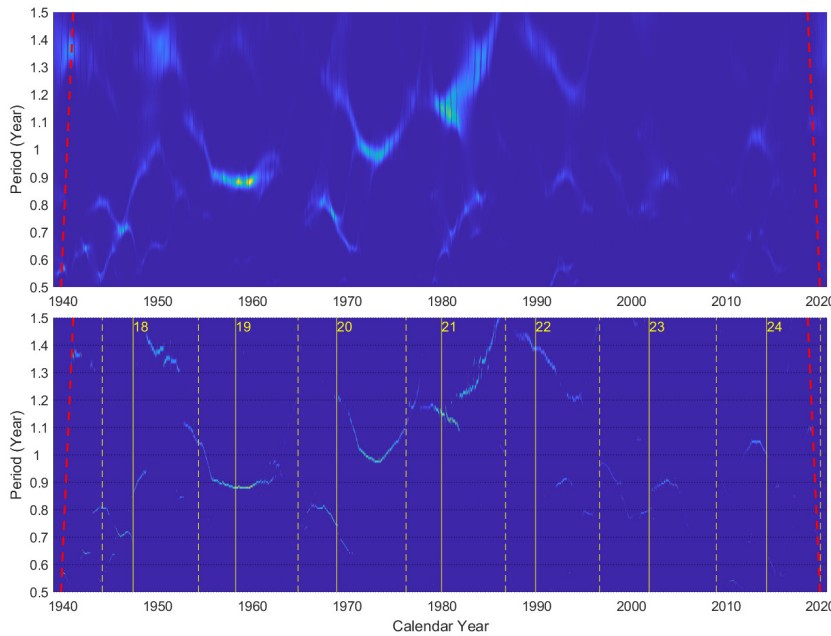

**Figure 11.** Same as Figure 6, but for periods between 0.5 and 1.5-yr.

### 3.3. Variation Pattern of Rieger Type Periods

Periodicities in the range from 120 to 200-day can be taken as the Rieger-type periodicities. The cause for the origin of Rieger-type periodicities may be a property of emerging flux rather than the total amount of flux present on the solar disk [41]. Carbonell and Ballester [43] suggested its association with the periodicity in emergence of magnetic flux through the photosphere. Figure 12 demonstrates the time–frequency plane of time scale ranges from 120 to 200-day along with the temporal variation profile of periods. One can know that the periods for such time scale also appear somewhat intermittently, and these periods were operating with the highest power in cycles 21 and 22, while the power is much lower in cycles 23 and 24. The average value of periods is 161.61-day with the standard deviation of 21.96-day.

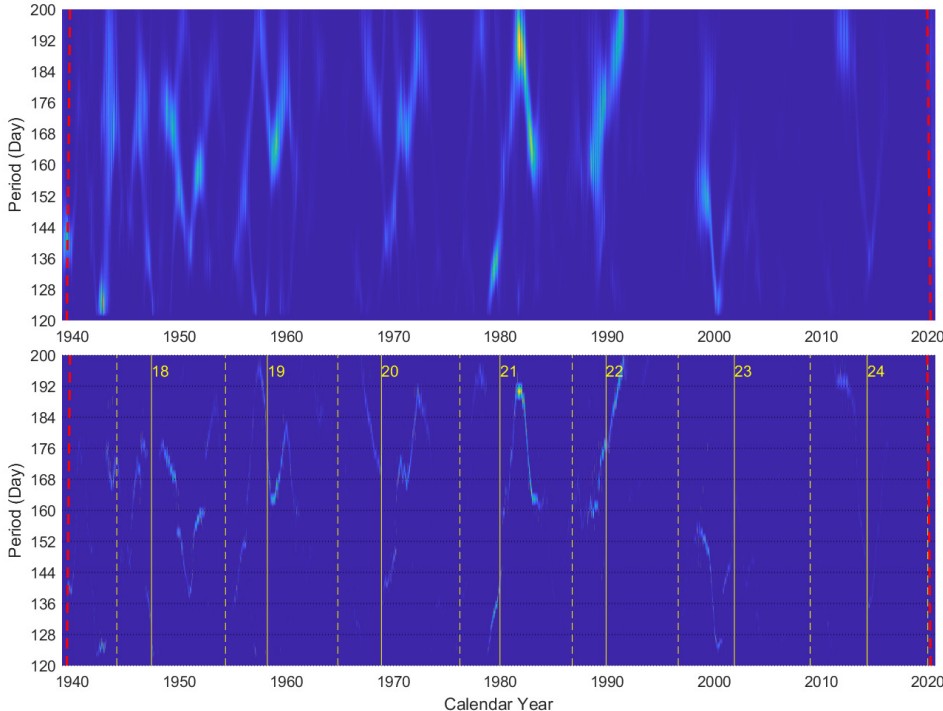

**Figure 12.** Same as Figure 6, but for periods between 120 and 200-day.

### 3.4. The Rotation Periods

The periods with their lengths close to 27-day are regarded as rotation periods. Deng et al. [27] studied the coronal rotation using a wavelet power spectrum, where they considered the rotation period between 26 and 38-day. Here, we consider the range from 20 to 40 days to detect the coronal rotation periods. The upper panel in Figure 13 presents the spectrum of MCI series with periods between 20 and 40-day. It can be seen that relatively higher power belts appear around 27 days without the appearance of any smoothly temporal variation profile. One can calculate that the average period is 28.71-day with the standard deviation of 4.24-day.

However, it can clearly be seen that the distribution of higher power is asymmetrical, namely, the power of its period greater than 27-day has more dispersion. Therefore, the quantile of distribution is more reasonable and robust to describe the primary feature in such case. The 25th, 50th (median), and 75th quantiles of the periods can be calculated as 26, 27.8, and 31-day, respectively. The lower panel in Figure 13 shows the temporal variation profile of the rotation period along with the 25th, 50th (median), and 75th quantile line. Here, the median of 27.8-day is a little bit different from 27.5-day obtained by Deng et al. [27]. This may be due to the difference of these methods, since we use the ConceFT method to calculate the median of 27.8-day according to the periods ranging from 20 to 40-day, while Deng et al. [27] obtained 27.5-day by global wavelet power spectra.

Ultimately, the results can be summarized in Table 1.

**Table 1.** The periodicities in the modified coronal index with variation patterns.

| Scale Type | Period | Variation Pattern | Significant Operating Duration |
|---|---|---|---|
| Schwabe cycle | $10.67 \pm 0.14$ yr | smooth | cycle 19–23 |
| mid-range | $8.07 \pm 0.38$ yr | smooth | cycle 21–22 |
| mid-range | $5.44 \pm 0.44$ yr | smooth | cycle 18–23 |
| mid-range | $3.42 \pm 0.24$ yr | local smooth | cycle 18, 19, 21 and 22 |
| mid-range | $1.01 \pm 0.24$ yr | local smooth | cycle 18–22 |
| QBOs | $2.3 \pm 0.42$ yr | local smooth | cycle 18–21 |
| Rieger type | $161.61 \pm 21.96$ day | local smooth | cycle 18–23 |
| Rotation | $27.8^{+3.2}_{-1.8}$ day | intermittent | cycle 18–24 |

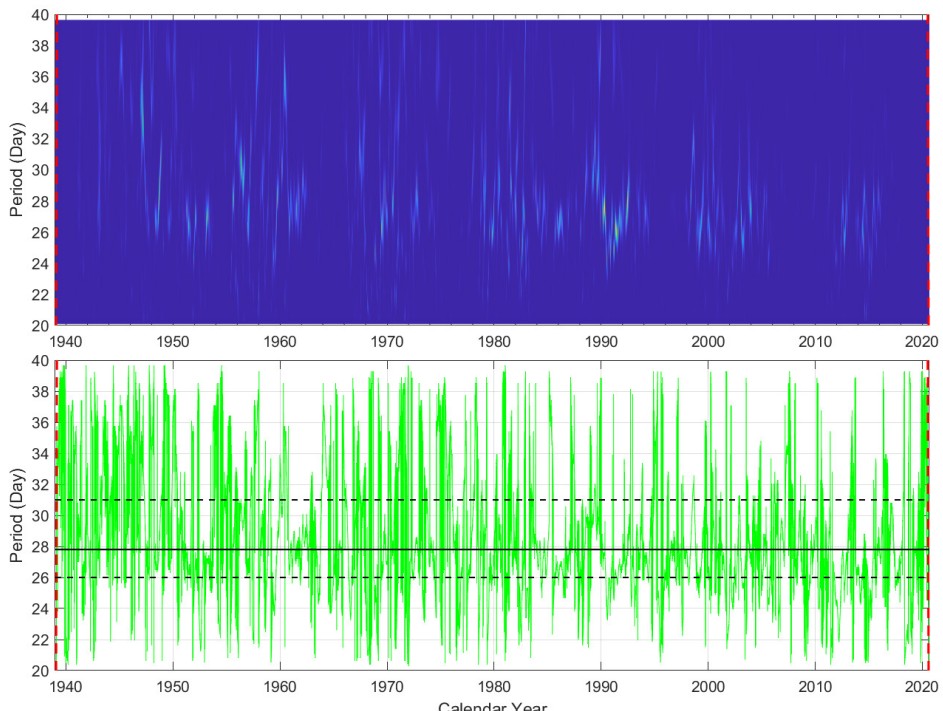

**Figure 13.** Same as Figure 6, but for periods between 20 and 40-day. In the lower panel, the solid black line presents the median of periods with a value of 27.8-day; the lower and upper dashed black lines indicate the 25th and 75th quantiles of the periods with the value of 26 and 31-day, respectively.

Here, we classify these periodic variations as three types according to their patterns:

(1) Smooth type, which means that a period with higher power varies relatively smoother during some certain time interval, such as in Figures 6–8;

(2) Local smooth type, it means that the period varies relatively smoother in some time intervals, but appears intermittently in other intervals, as is shown in Figures 9–12.

(3) Intermittent type, which means that the variation of the period appears intermittently in the entire time interval, as can be seen in Figure 13.

## 4. Conclusions

In this work, using the daily series of Modified Coronal Index (MCI) during a time interval from 1 January 1939 to 31 August 2020, periodicities in the coronal are investigated by the ConceFT method combined with the Lomb–Scargle periodogram. Compared to the traditional continuous wavelet technique, the ConceFT method is more powerful for concentrating the energy of a signal and is more capable of diminishing the noise. We focus

on the significant periods gained by the Lomb–Scargle periodogram and other meaningful periods to study the periodical variations in the corona. Our main conclusions are as follows:

(1) The Schwabe cycle, with its average length of 10.67-yr, is operating with the values between 10.5 and 11-yr during solar cycles from 19 to 23.

(2) For Rieger type periods, the average value of period length is 161.61-day with standard deviation of 21.96-day. The pattern exhibits somewhat intermittently; and it is operating with highest power in cycles 21 and 22, while the power is much lower in cycles 23 and 24.

(3) For rotation periods, the temporal variation exhibits a highly intermittent pattern in all solar cycles, but relatively higher power belts appear around 27-day as asymmetrical distribution with the 25th, 50th (median), and 75th quantile of 26, 27.8, and 31-day, respectively.

(4) For other mid-range periods, there are a total of five patterns of period variation. The first one is operating with its average period of 8.07-yr and standard deviation of 0.38-yr, which varies between 7.5 and 8.5-yr during solar cycles 21–22. The second pattern is operating as its period length declines from 5.8-yr to 5.5-yr in the duration from maximum times of solar cycle 18 to the maximum times of solar cycle 19; and duration from the maximum times of solar cycle 19 to maximum times of solar cycle 20, the period of its length increases from 5.5-yr up to 5.8-yr again. During cycle 21–23, such prominent period increases from 4.8-yr to 6.3-yr. The third pattern, with its average value of period length of 3.42-yr, is continuously operating from the start of cycle 21 to the end of cycle 22, while it appears intermittently during other time intervals. The fourth pattern, which may be related to the QBOs with an average period of 2.3-yr, appears much more intermittently during cycles 18 to 24 with higher energy during the time from the maximum times of cycle 18 to the same times of cycle 19. The fifth pattern with its average period of 1.01-yr is smoothly operating locally with higher power in cycles 19–22, while its power is much lower in other solar cycles.

**Author Contributions:** Conceptualization, Y.F.; Data curation, R.T., C.L., W.L. and X.T.; Formal analysis, R.T., Y.F. and C.L.; Funding acquisition, Y.F.; Investigation, W.L., X.T. and Z.W.; Methodology, R.T. and Y.F.; Software, C.L., W.L. and Z.W.; Supervision, Y.F.; Writing—original draft, R.T. and C.L.; Writing—review & editing, Y.F. All authors have read and agreed to the published version of the manuscript.

**Funding:** This research was funded by Yu Fei's research supported by the National Natural Science Foundation of China (NSFC) grant (No.11971421) and Yunling Scholar Research Fund of Yunnan Province (YNWR-YLXZ-2018-020).

**Data Availability Statement:** The data series of the modified coronal index is downloaded from the website of the Slovak Central Observatory in Hurbanovo (http://www.suh.sk/obs/vysl/MCI.htm, accessed on 15 June 2022). The series of sunspot numbers can be downloaded from the website http://www.sidc.be/silso/datafiles, accessed on 15 June 2022.

**Acknowledgments:** We would like to thank two anonymous referees for valuable comments and suggestions that helped to improve this manuscript significantly. We also thank to Torrence & Compo who provided the codes of wavelet transform analysis (https://atoc.colorado.edu/research/wavelets, accessed on 15 June 2022).

**Conflicts of Interest:** The authors declare no conflict of interest.

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
