# Peer review of "Periodic Variations of Solar Corona Index during 1939–2020"

_universe, doi:10.3390/universe8070375_

Round 1

Reviewer 1 Report

To improve this study, I recommend you the following suggestions.

In general, on several claimed periodicities, the statistical significances were not given.  You may refer the following paper and you could provide the statistical significance of these enhancements detected by your new wavelet packet. Then they will be more reliable, otherwise some of them may arise from noise. The paper entitled " A guide to Wavelet Analysis2 written by Torrence and Compo, published by Bulletin of the American Meteorlogocal Society.

Here are three main points that should be examined by the authors:

(1)  Why did not you try try the new method  to the sun spot numbers? Why did you select the coronal index? You should try your method also to the well-established sun spot numbers. and extend the discussions.

(2)  I found a serious discrepancy between the observed data and your analyzed data between 2000 and 2020. As you have shown in your paper, the standard CWT method suggests the prolongation of the periodicity during 2000 and 2020.  And this prediction well fits the observed and accepted results. But your newly choosed wavelet packet predicts the opposite way. How to explain this discrepancy between the observed result.

(3)  Finally, how to evaluate each excess claimed by you. What are the sigma of these excesses (confidence level) that you claim to have found by the mathematical method?  If you believe that they are real, you should demonstrate that they are sure and the data have the same tendency also in the sun spot numbers.

    By the simple Fourier analysis, we can find several periodicities as I attached this letter.

    Your figures from Figure 5 ,6,7 and 8 must have no scientific meaning. They should be removed from the text.

Reviewer 2 Report

This paper examines the periodic variation in the solar corona index over an 81 year period, but employing a novel transform method, ConceFT.

This paper is very well written - it has an excellent introduction, is clear, and the data is well presented. There are a few instances of incorrect wording (e.g. line 37, "pained" rather than "paid"), but these are minor.

There are a few points regarding the transform that require clarification/adding to the paper:

  1. When taking the CWT-based ConceFT approach, vertical striations appear in the data (figure 2, lower panel; figure 3-8, upper panels). Is this a side effect of the tapering method?  
  2. In Figure 2, towards the year 2020, the CWT shows an upward trend in period; the ConceFT shows a downward trend - can the authors explain this?
  3. In figure 2, is the same colorbar used for both plots?
  4. In general for the above three points, does the ConceFT have any limitations/disadvantages? Is there a loss/missing of any trends that are treated as "noise"?

Round 2

Reviewer 1 Report

The quality of the paper has been much improved and it has become easier to understand what you have discovered.  However, still some works remain, in order to get the credit what you have found.  Actually, it was wise to demonstrate using the Sun Spot Number that your method is correctly working in the analysis of the coronal index.  However in order to establish several periodicities, still some works leave.

First point: I am skeptical on the several periodicities what you claim, i. e, they might appear from higher order beat of the strong 11-year solar activity. 

For example, 10.27years /3= 3.4 years (you claim 3.27 years)

            10.27 years/2 = 5.1 years (you claim 5.45 years)

This matter is unavoidable when we make the data analysis by using Fourier expansion method..

Second point;

You have already presented in the (upper) Figure 3, there is COI effect.

Even several people try using different wavelet packet, however from the “corner effect”, nobody can escape. Even the ConceFT method, it would not be free from this effect.  By this reason, the 9-year periodicity during 1940-1955 in Figure 6 should not be discussed for scientific purpose. The same reason, during 1940-1950 of Figure 7 and probably 1940-1950 of Figure 8. However, the excess during 1975-1995 may have a scientific meaning and also during 1950-1960 of Figure 8.  Although, the latter may be induced by the beat of 11 years periodicity.

Since we cannot separate the systematic error from the statistical error, the numbers of 99.9999% in Table 1 has no meaning.  Therefore Table 1 should be removed. Figure 2 is enough.

To avoid this “edge effect”, I propose to use the Sun Spot Number again.  Why do not you use the SSN, for example, during 1900-2020?  Then you can avoid the edge effect on the duration of 1940 -1950.  If your discovery is true, the 9-year periodicity should appear in your analysis.

Minor comment: the word MCI is difficult to understand when I read and try to know Figures. I recommend you to write by using the full spelling in each figure caption, like” modified coronal index”.

The quality of the paper has been much improved and it has become easier to understand what you have discovered.  However, still some works remain, in order to get the credit what you have found.  Actually, it was wise to demonstrate using the Sun Spot Number that your method is correctly working in the analysis of the coronal index.  However in order to establish several periodicities, still some works leave.

First point: I am skeptical on the several periodicities what you claim, i. e, they might appear from higher order beat of the strong 11-year solar activity. 

For example, 10.27years /3= 3.4 years (you claim 3.27 years)

            10.27 years/2 = 5.1 years (you claim 5.45 years)

This matter is unavoidable when we make the data analysis by using Fourier expansion method..

Second point;

You have already presented in the (upper) Figure 3, there is COI effect.

Even several people try using different wavelet packet, however from the “corner effect”, nobody can escape. Even the ConceFT method, it would not be free from this effect.  By this reason, the 9-year periodicity during 1940-1955 in Figure 6 should not be discussed for scientific purpose. The same reason, during 1940-1950 of Figure 7 and probably 1940-1950 of Figure 8. However, the excess during 1975-1995 may have a scientific meaning and also during 1950-1960 of Figure 8.  Although, the latter may be induced by the beat of 11 years periodicity.

Since we cannot separate the systematic error from the statistical error, the numbers of 99.9999% in Table 1 has no meaning.  Therefore Table 1 should be removed. Figure 2 is enough.

To avoid this “edge effect”, I propose to use the Sun Spot Number again.  Why do not you use the SSN, for example, during 1900-2020?  Then you can avoid the edge effect on the duration of 1940 -1950.  If your discovery is true, the 9-year periodicity should appear in your analysis.

Minor comment: the word MCI is difficult to understand when I read and try to know Figures. I recommend you to write by using the full spelling in each figure caption, like” modified coronal index”.

v
